# Effects of Potassium-Containing Fertilizers on Sugar and Organic Acid Metabolism in Grape Fruits

**DOI:** 10.3390/ijms25052828

**Published:** 2024-02-29

**Authors:** Jin Wang, Yuhang Lu, Xuemei Zhang, Wenjie Hu, Lijin Lin, Qunxian Deng, Hui Xia, Dong Liang, Xiulan Lv

**Affiliations:** 1College of Horticulture, Sichuan Agricultural University, Chengdu 611130, China; 14224@sicau.edu.cn (J.W.); luyuhang@stu.sicau.edu.cn (Y.L.); 2022205015@stu.sicau.edu.cn (X.Z.); 14208@sicau.edu.cn (L.L.); dengqx@sicau.edu.cn (Q.D.); xiahui@sicau.edu.cn (H.X.); liangeast@sicau.edu.cn (D.L.); 2College of Science, Sichuan Agricultural University, Ya’an 625014, China; grapehu@stu.sicau.edu.cn

**Keywords:** fruits, potassium, quality, transcriptome analysis

## Abstract

To identify suitable potassium fertilizers for grape (*Vitis vinifera* L.) production and study their mechanism of action, the effects of four potassium-containing fertilizers (complex fertilizer, potassium nitrate, potassium sulfate, and potassium dihydrogen phosphate) on sugar and organic acid metabolism in grape fruits were investigated. Potassium-containing fertilizers increased the activity of sugar and organic acid metabolism-related enzymes at all stages of grape fruit development. During the later stages of fruit development, potassium-containing fertilizers increased the total soluble solid content and the sugar content of the different sugar fractions and decreased the titratable acid content and organic acid content of the different organic acid fractions. At the ripening stage of grape fruit, compared with the control, complex fertilizer, potassium nitrate, potassium sulfate, and potassium dihydrogen phosphate increased the total soluble solid content by 1.5, 1.2, 3.5, and 3.4 percentage points, decreased the titratable acid content by 0.09, 0.06, 0.18, and 0.17 percentage points, respectively, and also increased the total potassium content in grape fruits to a certain degree. Transcriptome analysis of the differentially expressed genes (DEGs) in the berries showed that applying potassium-containing fertilizers enriched the genes in pathways involved in fruit quality, namely, carbon metabolism, carbon fixation in photosynthetic organisms, glycolysis and gluconeogenesis, and fructose and mannose metabolism. Potassium-containing fertilizers affected the expression levels of genes regulating sugar metabolism and potassium ion uptake and transport. Overall, potassium-containing fertilizers can promote sugar accumulation and reduce acid accumulation in grape fruits, and potassium sulfate and potassium dihydrogen phosphate had the best effects among the fertilizers tested.

## 1. Introduction

The grape (*Vitis vinifera* L.) is a perennial vine fruit tree and one of the four major fruit trees in the world [1]. In 2021, the cultivation area of grapes in China was 783,000 ha, ranking third in the world, and the grape yield was 11.2 million tons, with the yield of table grapes accounting for 90% of the world’s production [2]. Grapes are potassium-loving fruit trees, and potassium fertilizers are more effective than nitrogen and phosphorus fertilizers in increasing the vitamin C and sugar content and decreasing the acid content in grape fruits [3]. However, in grape production, the heavy application of nitrogen and phosphorus fertilizers and the light application of potassium fertilizers are still common practices in some grape-growing regions, while in other instances, potassium fertilizers are not applied in a timely manner. These management practices result in a nutrient imbalance and low soil potassium concentration, which affects the quality of grape fruits [4,5,6]. Therefore, to further improve the quality of grape fruits, it is necessary to change the fertilization practices, notably by increasing the application of potassium fertilizers and balancing the nutrient content in the soil.

Potassium is an essential mineral that affects fruit quality and is recognized as a quality element [7]. Potassium can improve fruit quality by promoting the accumulation of assimilates, increasing the number of fruit cells, enlarging the cell volume of fruit tissues, and increasing the weight of single fruits and the soluble sugar and anthocyanin content [8,9,10,11]. Applying potassium fertilizers can increase fruit sucrose phosphate synthase (SPS) activity and promote the accumulation of sucrose, which is then converted into fructose and glucose [12,13]. Potassium fertilizer application can also activate sugar metabolism-related enzymes, such as phosphofructokinase and starch synthetase, which can affect the sugar metabolism process in fruits [14]. In figs (*Ficus carica* Linn.), applying potassium fertilizers increases acidic invertase (AI), sucrose synthase (SS, degradation direction), and neutral invertase (NI) activity, promotes fructose and glucose accumulation, increases α-amylase and β-amylase activity, and consequently, facilitates the conversion of starch to soluble sugar in the fruit during the early and late stages of development [15]. In jujube (*Ziziphus jujuba* Mill.), applying potassium fertilizers increases the SPS, SS, AI, and NI activity in fruits, and these enzymes mainly respond to potassium fertilizers during the fruit expansion and sclerotization phase [16]. In grapes, applying potassium fertilizers can up-regulate the expression levels of *AI*, *SS*, and *SPS* in the fruits and increase the total soluble sugar, glucose, and fructose content in the fruits [17,18]. In addition, the K^+^ accumulated in the fruits during fruit ripening neutralizes organic acids, which, in turn, reduces fruit acidity in warm climates [19]. Concurrently, potassium uptake by the fruits increases the release of malic acid, which is metabolized in the cytoplasm [20]. Applying potassium fertilizers inhibits the malate dehydrogenase (MDH) and phosphoenolpyruvate carboxylase (PEPC) activity involved in malic acid synthesis and increases the phosphoenolpyruvate carboxykinase (PEPCK) and malic enzyme (ME) activity involved in malic acid degradation, thus reducing the accumulation of organic acids in the fruits [21,22]. For example, foliar spraying of potassium dihydrogen phosphate has been shown to reduce the tartaric, malic, citric, and manganic acid content in grape fruits [17,18]. Therefore, applying potassium fertilizers has the effect of increasing the sugar content and decreasing the acidity in fruits.

Grape fruits require enormous amounts of potassium and the amount of potassium fertilizers they require increases over the course of fruit development [23]. Thus, applying appropriate amounts of potassium fertilizers during the development of grape fruits can improve fruit yield and quality [24]. However, although there is a wide variety of potassium fertilizer formulations available for agricultural production [25,26], the action mechanisms of potassium fertilizers on grape fruits still have many questions and need to be explored. The mechanisms behind the sugar-increasing and acidity-reducing responses after potassium fertilizer applications in grape fruits remain unknown. In this study, the effects of four potassium-containing fertilizers (complex fertilizer, potassium nitrate, potassium sulfate, and potassium dihydrogen phosphate) on sugar and organic acid metabolism in grape fruits were investigated. The aims of this study were to (1) screen for suitable potassium fertilizers in grape production and (2) decipher the mechanisms by which potassium fertilizers affect the sugar and organic acid metabolism in grape fruits.

## 2. Results

### 2.1. Sugar Content in the Grape Fruits

As the number of days after treatment increased, the total soluble solids (TSSs), total soluble sugar, reducing sugar, fructose, sucrose, and glucose content increased in the grape fruits (Figure 1A–F). Potassium sulfate (PS) and potassium dihydrogen phosphate (PDP) increased the TSS content to some extent on all the sampled days after treatment, while complex fertilizer (CF) and potassium nitrate (PN) only increased on the 28th, 35th, and 42nd days after treatment. CF, PN, PS, and PDP increased the total soluble sugar, reducing sugar, fructose, sucrose, and glucose content on all days following treatment, to some extent. On the 42nd day after treatment, the order of the TSS and total soluble sugar content was PDP ≈ PS > PN ≈ CF > control (CK). CK, CF, PN, PS, and PDP increased the TSS content by 1.5, 1.2, 3.5, and 3.4 percentage points and the total soluble sugar content by 19.4%, 17.0%, 26.3%, and 27.8%, respectively, on the 42nd day after treatment.

### 2.2. Sugar Metabolism-Related Enzyme Activity in Grape Fruits

According to the changes in sugar content in the grape fruits, the 14th, 28th, and 42nd days after treatment were chosen for the sugar metabolism-related enzyme activity measurement in the grape fruits (Figure 2A–D). CF, PN, PS, and PDP increased the activity of SS, SPS, AI, and NI in the grape fruits on the 14th, 28th, and 42nd days after treatment. On the 42nd day after treatment, CF, PN, PS, and PDP increased SS activity by 73.8%, 68.1%, 89.6%, and 94.1%, SPS activity by 85.4%, 83.5%, 95.6%, and 110.1%, NI activity by 44.1%, 33.5%, 59.1%, and 79.2%, and SPS activity by 41.8%, 30.1%, 47.9%, and 63.9%, respectively, compared to CK.

### 2.3. Organic Acid Content in Grape Fruits

As the number of days after treatment increased, the titratable acid (TA), oxalic acid, tartaric acid, L-malic acid, D-malic acid, citric acid, and succinic acid content in the grape fruits increased until the 14th day and decreased thereafter (Figure 3A,C–H). The organic acid content of the different compounds also peaked on the 14th day after treatment. CF, PN, PS, and PDP decreased the TSS content to some extent on all days after treatment. On the 14th (and later days) after treatment, CF, PN, PS, and PDP decreased the oxalic acid, tartaric acid, L-malic acid, D-malic acid, citric acid, and succinic acid content to some extent. On the 42nd day after treatment, the order of TA content was CK > CF ≈ PN > PS ≈ PDP. CK, CF, PN, PS, and PDP decreased the TA content by 0.09, 0.06, 0.18, and 0.17 percentage points, respectively, on the 42nd day after treatment. As the number of days after treatment increased, the TSS/TA ratio of the grape fruits also displayed an increasing trend (Figure 3B). CF, PN, PS, and PDP increased the TSS/TA ratio to some extent on all days after treatment. On the 42nd day after treatment, the order of the TSS/TA ratio was PS ≈ PDP > CF ≈ PN > CK.

### 2.4. Organic Acid Metabolism-Related Enzyme Activity in Grape Fruits

The organic acid metabolism-related enzyme activity in the grape fruits was measured on the 14th, 28th, and 42nd days after treatment because of the changes observed in the organic acid content in grape fruits on those days (Figure 4A–F). CF, PN, PS, and PDP increased the activity of citrate synthetase (CS), PEPC, ME, MDH, aconitase (ACO), and isocitrate dehydrogenase (ICD) in the grape fruits on the 14th, 28th, and 42nd days after treatment. On the 42nd day after treatment, CF, PN, PS, and PDP increased CS activity by 21.2%, 19.1%, 51.2%, and 29.1%, PEPC activity by 46.9%, 37.6%, 68.0%, 62.2%, ME activity by 114.1%, 94.5%, 144.8%, and 135.4%, MDH activity by 31.6%, 24.7%, 48.3%, and 36.3%, ACO activity by 58.5%, 47.5%, 95.2%, and 82.4%, and ICD activity by 47.8%, 46.0%, 71.8%, and 48.9%, respectively, compared to CK.

### 2.5. Total Potassium Content in Grape Fruits

The total potassium content in grape fruits on the 42nd day after treatment was determined (Figure 5). CF, PN, PS, and PDP increased the total potassium content in the skin and pulp of grape fruits to a certain degree. PS and PDP significantly increased the total potassium content in skin and pulp. CF and PN had no significant effects on the total potassium content in the skin. CF significantly increased the total potassium content in pulp, while PN had no significant effects on this parameter.

### 2.6. DEG Analysis following Potassium-Containing Fertilizer Treatment

To further investigate the effects of potassium-containing fertilizers on sugar and organic acid metabolism in the grape fruits, transcriptome data from the berries sampled on the 14th, 28th, and 42nd days after treatment were analyzed. A total of 45 samples (C represents the 14th day, E represents the 28th day, and G represents the 42nd day) were sequenced on the Illumina platform. The sequencing results revealed that the GC content was greater than 46%, the Q30 score was greater than 91%, and the Q20 score was greater than 96% (Table 1), thereby implying a high sequencing and library preparation quality of the samples. The principal component analysis revealed a high correlation between all samples (Figure 6). The transcriptome data from the 14th, 28th, and 42nd days after treatment revealed a total of 34,678, 29,459, and 53,648 DEGs, respectively, after applying potassium-containing fertilizers (Figure 7). On the 14th day after treatment, the highest number of DEGs was found in CPDP vs. CCK, with a total of 6147 DEGs, including 2799 upregulated and 3348 downregulated genes. The second highest number of DEGs was found in CPS vs. CCK, with 4679 DEGs, which included 2348 upregulated and 2331 downregulated genes. On the 28th day after treatment, EPDP vs. ECK had the highest number of 4406 DEGs, which included 2181 upregulated and 2225 downregulated genes, followed by EPS vs. ECK with 1556 DEGs, which included 710 upregulated and 846 downregulated genes. On the 42nd day after treatment, GPS vs. GCK had the highest number of 7578 DEGs, which included 3667 upregulated and 3911 downregulated genes, followed by GPN vs. GCK, with a total of 6714 DEGs, including 3512 upregulated and 3202 downregulated genes.

The DEGs of three data sets from different control groups were compared using Venn diagrams (Figure 8A–D). The results showed that 937, 390, and 2057 genes were differentially enriched in the treatment group compared to the control on the 14th, 28th, and 42nd days after treatment, respectively (Figure 8A–C). On the 14th and 28th days after treatment, 49 DEGs were shared by both the treatment group and the control (Figure 8D). On the 14th and 28th days after treatment, there were 224 DEGs in common between the treatment group and the control. On the 28th and 42nd days after treatment, the treatment group shared 106 DEGs with the control, of which 19 genes were differentially enriched in all three periods after treatment.

### 2.7. Functional Classification of DEGs

The GO enrichment analysis revealed that the DEGs from the 14th day after treatment were involved in biological processes, cellular components, and molecular functions (Figure 9). Among the biological processes, the DEGs of CF vs. CK were mainly involved in carbohydrate biosynthesis and metabolism, polysaccharide metabolism, and pyruvate biosynthesis and metabolism, the DEGs of PN vs. CK in amino sugar synthesis and metabolism and cell wall macromolecule biosynthesis and metabolic processes, the DEGs of PS vs. CK in carbohydrate metabolism, ATP biosynthesis and catabolism, and organic matter degradation, and the DEGs of PDP vs. CK in organic acid biosynthesis, polysaccharide metabolism, and carbohydrate metabolism. Among the cellular components, the DEGs of CF vs. CK were mainly involved in cysts, photosynthetic membranes, and photosystems, the DEGs of PN vs. CK in cell walls and extracellular region, the DEGs of PS vs. CK in cell wall and extracellular body, and the DEGs of PDP vs. CK in cysts, photosynthetic membrane, photosystems, and extracellular body. Among the molecular functions, the DEGs of CF vs. CK were mainly involved in calcium binding, hydrolase activity, and glycosyltransferase activity, the DEGs of PN vs. CK and PDP vs. CK in hydrolase activity and glycosyltransferase activity, and the DEGs of PS vs. CK in hydrolase activity and pyrophosphatase activity. On the 28th day after treatment, the GO enrichment analysis revealed that the DEGs of CF vs. CK and PS vs. CK did not include significantly enriched genes involved in biological processes, cellular components, and molecular functions (Figure 10). The DEGs of PN vs. CK did not reveal significantly enriched genes involved in biological processes, but these genes were mainly involved with the cytoplasm, the intracellular non-membrane organelles of cellular components, the structural composition of ribosomes, and the structural molecular activity of molecular functions. The DEGs of PDP vs. CK were also not significantly enriched in genes involved in biological processes, but they were mainly involved with the vesicle, photosynthetic membrane, photosystem, extracorporeal body of cellular components, hydrolase activity, and glycosyltransferase activity. On the 42nd day after treatment, among the biological processes, the DEGs of CF vs. CK were mainly involved in glucan synthesis and metabolism, carbohydrate synthesis and catabolism, polysaccharide synthesis and catabolism, and pyruvate biosynthesis and metabolism, the DEGs of PN vs. CK in translated proteins, peptide biosynthesis and metabolism, amide biosynthesis and metabolism, carbohydrate metabolism, and metal ion transport, and the DEGs of PDP vs. CK in the translated proteins, peptide biosynthesis and metabolism, amide biosynthesis and metabolism, carbohydrate metabolism, and glycolysis (Figure 11). The DEGs of PS vs. CK did not include genes that were involved in biological processes. Among the cellular components, the DEGs of CF vs. CK were mainly involved in the cysts, membrane protein complexes, and photosystems, the DEGs of PN vs. CK in the ribosomes, ribonucleoprotein complexes, and membrane-less organelles, the DEGs of PS vs. CK in the endosomes, photosystems, and photosynthetic membranes, and the DEGs of PDP vs. CK in the ribosomes. Among the molecular functions, the DEGs of CF vs. CK were mainly involved in calcium binding, hydrolase activity, transporter activity, and glycosyltransferase activity, the DEGs of PN vs. CK in the structural composition of ribosomes, hydrolytic enzyme activity, transmembrane transporter activity, and nucleoside triphosphate reductase activity, the DEGs of PS vs. CK in acyltransferase activity and transporter activity, and the DEGs of PDP vs. CK in the structural composition of ribosomes, hydrolytic enzyme activity, transmembrane transporter activity, and structural molecule activity.

According to KEGG pathway enrichment analysis, the DEGs belonging to the 20 most important KEGG pathways were mainly involved in phytohormone signaling, amino acid biosynthesis, carbon metabolism, ribosome, and glycolysis and gluconeogenesis (Figure 12, Figure 13 and Figure 14, Table 2). The types and number of metabolic pathways enriched in response to potassium-containing fertilizer treatments were lower on the 28th day after treatment than on the 14th and 42nd days after treatment. On the 14th day after treatment, the most highly enriched DEG pathway was in PDP vs. CK (Figure 12). The DEGs of the four fertilizer treatments, when compared with CK, were involved in carbon fixation in photosynthetic organisms. The DEGs of CF, PS, and PDP vs. CK were involved in glycolysis, gluconeogenesis, and carbon metabolism. The DEGs of PS and PDP vs. CK were also involved in fructose and mannose metabolism, and the DEGs of PN vs. CK were involved in ribosomes and phytohormone signaling. On the 28th day after treatment, the DEGs of CF vs. CK were involved in phenylpropane biosynthesis, phytohormone signaling, galactose metabolism, plant–pathogen interactions, and plant–pathogen interactions (Figure 13). The DEGs of PN vs. CK were involved in galactose metabolism and endoplasmic reticulum protein processing, and the DEGs of PS vs. CK were also involved in endoplasmic reticulum protein processing. The DEGs of PDP vs. CK were involved in photosynthesis. On the 42nd day after treatment, the DEGs of CK, PS, and PDP vs. CK were involved in a larger variety and number of metabolic pathways than PN vs. CK (Figure 14). The carbon fixation pathway by photosynthetic organisms was enriched in treatments vs. CK and had the highest correlation among all the pathways. The carotenoid biosynthesis and fructose biosynthesis pathways were also enriched in the PS, PN, and PDP treatments vs. CK. In addition, 11 DEGs related to sugar metabolism and potassium transport were identified in the transcriptome sequencing data (Table 3). The expression levels of these 11 genes were subsequently verified with qRT-PCR.

### 2.8. qRT-PCR Analysis of DEGs

The 11 DEGs related to sugar metabolism and potassium transport were successfully verified with qRT-PCR (Figure 15). Throughout the three developmental periods of the grape berries, the expression levels of the DEGs related to potassium transport protein biosynthesis and sugar metabolism were increasingly upregulated, suggesting that applying potassium-containing fertilizers promoted the accumulation of potassium ions and sugar in the grape berries. The upregulated expression levels of *PK*, *PFK*, *GolS1*, *SS*, and *PHT14* promoted the accumulation and transport of soluble sugars on the 14th, 28th, and 42nd days after treatment. The expression levels of *SOR*, *AKT2*, and *OCT3* were higher than in CK on the 14th and 28th days after treatment, and those of *CNGC4*, *NFXL1*, and *PT6* were higher than in CK on the 28th and 42nd days after treatment. The highest expression levels of these genes were observed in PS, followed by PDP. The expression levels of *PT6* were higher than in CK on the 14th, 28th, and 42nd days after treatment, with PS and PDP being the highest. The expression levels of *SOR* in the four treatments were higher than in CK on the 14th and 28th days after treatment and lower than in CK on the 42nd day after treatment.

## 3. Discussion

### 3.1. Effects of Potassium-Containing Fertilizers on the Sugar Content in Fruits

The sugar and organic acid contents in fruits are important determinants of fruit flavor [27]. Grapes are hexose-accumulating fruits, and fructose, glucose, and sucrose are the main components of soluble sugars in grape fruits; in addition, fructose plays an important role in the fruits’ sweetness [28]. Applying potassium fertilizers to the soil can increase the sugar content in grape fruits [17,29]. In this study, a gradually increasing trend was observed in the sugar content of the different sugar fractions over time after treatment, and the sucrose, glucose, fructose, and total soluble sugar content in the grape fruits increased seven days after applying complex fertilizer, potassium nitrate, potassium sulfate, and potassium dihydrogen phosphate. Applying potassium sulfate and potassium dihydrogen phosphate improved glucose accumulation more than applying complex fertilizer and potassium nitrate. The results from the present study are consistent with those from previous studies on grapes and tomatoes [18,30]. The reason may be due to the increased demand for potassium during the later stages of fruit growth and development, and the application of potassium-containing fertilizers improves the nutrient uptake process in the fruits, which, in turn, improves sugar accumulation [31]. So, potassium sulfate and potassium dihydrogen phosphate are recommended as suitable potassium fertilizers for increasing sugar accumulation in grape production.

### 3.2. Effects of Potassium-Containing Fertilizers on the Activity of Sugar Metabolism-Related Enzymes in Fruits

NI and AI can break down sucrose into glucose and fructose and regulate the sucrose transport out of the phloem; SPS is the key rate-limiting enzyme for sucrose biosynthesis, and its activity level has a direct impact on sucrose accumulation [32,33]. Potassium fertilizers can increase the activity of SPS, AI, NI, and SS in fruits by regulating the expression levels of sugar metabolism-related genes [30,34,35,36]. In this study, SS, SPS, AI, and NI activity increased in the grape fruits on the 14th, 28th, and 42nd days after applying potassium-containing fertilizers, and the activity of these sugar metabolism-related enzymes was highest after applying potassium dihydrogen phosphate and second highest after applying potassium sulfate. The activity levels of the sugar metabolism-related enzymes after applying potassium-containing fertilizers were consistent with the changes observed in the content of different sugar fractions, and this may be due to the fact that, during the late stage of fruit maturation, the nutrients in the tree shift from nitrogen to sugar, and potassium ions participate in the various processes of sugar transport, metabolism, biosynthesis, and accumulation [37]. Applying potassium-containing fertilizers promotes higher enzyme activity during sucrose synthesis and degradation and sugar accumulation in fruits. The differences observed in the activity levels of sugar metabolism-related enzymes when different potassium-containing fertilizers were applied may be due to the differences in the rates at which plants can take up the different forms of potassium, given the different ions accompanying the fertilizer formulations [3,7].

### 3.3. Effects of Potassium-Containing Fertilizers on the Organic Acid Content in Fruits

Organic acids are important factors that affect the sensory quality of fruits, and potassium plays an important role in acid metabolism in fruits [38]. The higher potassium accumulation in grapes causes the hydrogen ions from organic acids to be exchanged with potassium ions, thereby resulting in lower concentrations of free acids, especially tartaric acid, in the fruits; meanwhile, high concentrations of potassium inhibit the transfer and degradation of malic acid, thereby resulting in the accumulation of malic acid in fruits [39]. In grapes, applying potassium fertilizers reduces the tartaric and malic acid content in table grape fruits but increases their content in wine grape fruits [29]. In this study, the different organic acid fractions in grape fruits tended to increase and then decrease over the course of grape berry development. The oxalic, tartaric, L-malic, D-malic, citric, and succinic acids content in the grape fruits decreased to different degrees when complex fertilizer, potassium nitrate, potassium sulfate, and potassium dihydrogen phosphate were applied compared to the control, and the organic acid content was lower when potassium sulfate and potassium dihydrogen phosphate were applied. This was due to the fact that potassium-containing fertilizers promoted potassium ion accumulation in grape fruits. The higher potassium ion content reacts with the organic acids to form salts, thereby resulting in a decrease in organic acid content and suggesting that potassium fertilizers can play a role in lowering fruit acidity [39]. These results differ from the previous study on grapes [29] and may be related to the different grape species used in the two studies. So, potassium sulfate and potassium dihydrogen phosphate are recommended as suitable potassium fertilizers for reducing the organic acid accumulation in grape production.

### 3.4. Effects of Potassium-Containing Fertilizers on the Activity of Organic Acid Metabolism-Related Enzymes in Fruits

During fruit development, PEPC and MDH are two key enzymes that are involved in malic acid synthesis, and ME plays a similarly crucial role in malic acid degradation [40]. CS, ICD, and ACO are the key enzymes in the plant’s tricarboxylic acid (TCA) cycle and are involved in both the synthesis and degradation of citric acid [41]. Applying potassium fertilizers increases CS and PEPC activity in navel oranges, but they do not affect ICD and mitochondrial ACO activity [35]. In apples, applying potassium fertilizers decreases MDH and PEPC activity in the malic acid synthesis pathway and increases phosphoenolpyruvate kinase and ME activity in the malic acid degradation pathway [22]. In this study, PEPC, MDH, CS, and ACO activity in the grape fruits increased after applying potassium-containing fertilizers, thereby indicating that potassium-containing fertilizers can promote malic and citric acid synthesis in fruits to a certain extent. At the same time, ME and ICD activity also increased after applying potassium-containing fertilizers. The activity of organic acid metabolism-related enzymes in grape fruits was higher after applying potassium sulfate and potassium dihydrogen phosphate than complex fertilizer and potassium nitrate, and the concentration of the organic acid fractions in fruits was negatively correlated to that of the organic acid metabolism-related enzymes. This indicates that applying potassium-containing fertilizers could promote both the synthesis and the degradation of organic acids and that the effect of degradation was stronger than synthesis.

### 3.5. Effects of Potassium-Containing Fertilizers on the Sugar and Organic Acid Metabolism Pathways

Fruit sugar and organic acid metabolism are regulated by multiple genes, and the genes in these metabolic pathways are usually regulated in a coordinated fashion [42,43]. Potassium promotes sugar accumulation and storage in fruits by upregulating the expression levels of genes related to the glycolysis and fructose metabolism pathways. *PK* is the rate-limiting enzyme in glycolysis and regulates C/N metabolism [44]. *PFK* is also one of the key enzymes in the glycolysis process [14]. In this study, the expression levels of *PFK* and *PK* were upregulated on the 14th, 28th, and 42nd days after treatment, and *PFK* and *PK* were enriched in the glycolysis and gluconeogenesis pathways and the fructose and mannose metabolism pathways, thereby suggesting that 6-phospho-fructose and phosphoenolpyruvate are being metabolized.

Potassium can increase nitrate reductase activity, which, in turn, improves ammonia assimilation, promotes chlorophyll synthesis, and increases the net photosynthetic rate in plants [45,46]. Carbohydrates are an important by-product of photosynthesis and are the main source of energy for plants [47]. In this study, the DEGs identified when comparing potassium-containing fertilizer treatments with the control in the grape fruits were involved in porphyrin and chlorophyll metabolism, carbon fixation in photosynthetic organisms, and carbon metabolism pathways. This suggests that applying potassium-containing fertilizers can improve photosynthesis in grapes and, consequently, help synthesize more carbohydrates and promote sugar accumulation in fruits. In addition, sugar molecules must cross organelle membranes when entering chloroplasts [48], vesicles [49], and the Golgi apparatus [50]. The movement of sugar molecules across membranes, as well as during their transport and storage, involves several transporter proteins; more specifically, the proteins belonging to the MST subfamily *ERD6* or *ESL1* are involved in intravesicular sugar molecule transport [51]. The transcriptome and qRT-PCR analyses revealed that *PHT14*, *GolS1*, and *SS* were upregulated after applying potassium-containing fertilizers, which, in turn, promoted sucrose synthesis and increased the glucose and fructose content in fruits. These results are consistent with similar previous studies [52].

Potassium ion transport in cells mainly relies on either potassium ion transporter proteins or potassium ion transporter carriers, and the expression levels of the potassium uptake-related genes can directly reflect the degree to which potassium ion uptake and transport are active in grapevine plants [53]. Potassium fertilizer treatments, regardless of the concentration levels, can improve grape fruit quality by upregulating the potassium transporter protein genes (three *HAK*/*KUP*/*KT* genes) and sugar metabolism genes (*HT*, *SPS*, and *PK*) and, consequently, promoting the accumulation of potassium ions and sugars [18]. In loquats, the expression levels of the two-pore potassium channel protein gene *EjTP3* in the leaves increase with an increasing concentration of applied potassium fertilizers [54]. In *Arabidopsis*, the CNGC channel *AtCNGC2* has a high affinity for potassium ions [55], and *AtCNGC1* correlates with the potassium content in the aboveground part of the plant [56]. The Shaker K^+^ channel *AKT2* is mainly expressed in the thin-walled tissues of the xylem and phloem and is capable of transporting potassium ions in both directions [57]. *SOR* is an outwardly rectifying potassium ion channel gene in the Shaker family [53]. In this study, applying potassium-containing fertilizers upregulated the relative expression levels of the potassium ion uptake and transport-related genes *SOR*, *AKT2*, and *OCT3* on the 14th and 28th days after treatment and upregulated the expression levels of *CNGC4*, *NFXL1*, and *PT6* on the 28th and 42nd days after treatment, which was highly consistent with observed changes in the physiological indexes of the grapes. However, further investigation is required to identify the key genes responsible for coding the potassium transport proteins in fruits and to understand the mechanisms by which the expression of related proteins is regulated in plants.

## 4. Materials and Methods

### 4.1. Experimental Site and Materials

The experimental site was located in the vineyard at the Modern Agricultural Research and Development Base of Sichuan Agricultural University (33°33′ N, 103°38′ E), Chengdu, China. The study area belongs to the subtropical humid monsoon climate zone, with an average annual temperature of 15.9 °C, and an average annual rainfall of 1012.4 mm. The grapes used in this experiment were 8-year-old ‘Shine Muscat’, with rootstock ‘Beta’. Per hectare, 2250 grapes were planted, and the grape yield was maintained at 22.50–25.25 t/ha. The grapes were cultivated under a steel-framed rain shelter with a horizontal trellis. The top of the rain shelter was covered with a polyethylene dripless film with a thickness of 0.08 mm and a light transmittance of 95%.

The vineyard was grown on inceptisol soil with the following basic physical and chemical properties in the 0–40 cm layer: pH 7.78, organic matter content 23.22 g/kg, total nitrogen content 1.22 g/kg, total phosphorus content 12.55 g/kg, total potassium content 15.44 g/kg, alkali-hydrolyzable N content 112.85 mg/kg, available phosphorus content 58.57 mg/kg, and available potassium content 85.64 mg/kg. The soil properties were determined according to the methods by Bao (2000) [58].

The complex fertilizer was produced by Longmang Dadi Agricultural Co., Ltd. (Deyang, China) with total nutrients ≥ 45% (N:P_2_O_5_:K_2_O = 15:15:15). Potassium nitrate was produced by Huichuang Agricultural Development (Yunnan) Co., Ltd. (Kunming, China) with total nutrients ≥ 59.5% (N:K_2_O = 13.5:46.0). Potassium sulfate was produced by Zhonghua Chemical Fertilizer Co., Ltd. (Weihai, China) with K_2_O ≥ 52%. Potassium dihydrogen phosphate was produced by Sichuan Jiuhua Chemical Co., Ltd. (Yibin, China) with K_2_O ≥ 34% and P_2_O_5_ ≥ 52%.

### 4.2. Experimental Design

The experiment was conducted from July to August 2021, and uniformly growing grapevines were selected for the study. In accordance with the results from our previous study [59], 429 kg/ha K_2_O of potassium (191 g K_2_O per plant) was applied to the field, and this amount corresponded to 2862 kg/ha of complex fertilizer, 918 kg/ha of potassium nitrate, 825 kg/ha of potassium sulfate, and 1242 kg/ha of potassium dihydrogen phosphate. The corresponding amount of fertilizer that a single plant received was as follows: 1272 g of complex fertilizer, 408 g of potassium nitrate, 366 g of potassium sulfate, and 552 g of potassium dihydrogen phosphate. The experiment was set up with five treatments: no potassium fertilizer (control, CK), complex fertilizer (CF), potassium nitrate (PN), potassium sulfate (PS), and potassium dihydrogen phosphate (PDP). Each treatment was replicated three times, with five grapevines included in each replicate and a total of 75 grapevines. Potassium-containing fertilizers were divided equally into three applications according to the total amount and applied to each hole dug 20–30 cm from either side of the tree. The first application of potassium-containing fertilizers was conducted 60 days after flowering and then every seven days thereafter. The potassium-containing fertilizers were applied and followed by irrigation to fully dissolve the fertilizers. The other management practices of the grapevines were the same as in the standard ‘Shine Muscat’ grape production.

### 4.3. Sampling

Before the first sampling, each bunch of grape fruits with consistent development was labeled for sampling. Sampling was conducted a total of seven times, starting on the date of the first treatment and then every seven days until the grape fruits were fully ripe. At the time of sampling, 4–6 fruits were collected from the top, middle, and bottom of each bunch, and 50 fruits were taken from each replicate, placed in an ice box, and brought back to the laboratory to determine the visual quality, the total soluble solids (TSS) content, and the titratable acid (TA) content. The skins were separated from the rest of the fruits, and both parts were chopped, placed in liquid nitrogen, and then stored at −80 °C until ready to be used to determine the sugar and organic acid fractions content, the activity of sugar- and organic acid-related metabolism enzymes, total potassium content, transcriptome sequencing, and gene expression analysis.

### 4.4. Sugar and Organic Acid Content Measurement

The TSS content was determined using a hand-held refractometer with three biological repetitions. The TA content was determined using the sodium hydroxide titration method [60] with three biological repetitions. The TSS/TA ratio was then calculated as the TSS content/TA content.

The concentrations of the different sugar fractions (total soluble sugar, reducing sugar, fructose, sucrose, and glucose) were determined using a high-performance liquid chromatography (HPLC; Agilent 1260 system; Agilent Technologies, Santa Clara, CA, USA) with three biological repetitions. The chromatographic conditions for measuring the sugars were as follows: Innoval NH_2_ column (250 mm × 4.6 mm, 5 μm; Agela Technologies, Shanghai, China), mobile phase composed of acetonitrile and water (80:20, *v*/*v*), differential detector with a flow rate of 1 mL/min, a detection cell temperature of 40 °C, a column temperature of 25 °C, and an injection volume of 10 μL [61,62].

The organic acid fractions (oxalic, tartaric, L-malic, D-malic, citric, and succinic acids) content with three biological repetitions were also determined with an HPLC. The chromatographic conditions were: MZ PerfectSil Target HD C18 column (250 mm × 4.6 mm, 5 μm; MZ Analysentechnik, Mainz, Germany), the mobile phase composed of methanol and 0.2% phosphoric acid (3:97, *v*/*v*), flow rate of 0.8 mL/min, column temperature of 25 °C, detection wavelength of 210 nm, and an injection volume of 20 μL [61,62].

### 4.5. Determination of the Sugar and Organic Acid Metabolism-Related Enzyme Activity

The sugar and organic acid content results were used to select the 14th, 28th, and 42nd days after the first application of potassium-containing fertilizers as time points from which to measure the sugar and organic acid metabolism-related enzyme activity. The activities of the sugar metabolism-related enzymes [sucrose synthase (SS), sucrose phosphate synthase (SPS), acidic invertase (AI), and neutral invertase (NI)] and organic acid metabolism-related enzymes [phosphoenolpyruvate carboxykinase (PEPCK), malic enzyme (ME), malate dehydrogenase (MDH), citrate synthase (CS), isocitrate dehydrogenase (ICD), and aconitase (ACO)] with three biological repetitions were determined using enzyme-linked immunosorbent assay (ELISA) kits (Jiangsu Enzyme Immunity Industry Co., Ltd., Yancheng, China) according to the manufacturer’s instructions.

### 4.6. Determination of the Total Potassium Content

The berry skin and pulp samples of the 42nd days after the first application of potassium-containing fertilizers were used to determine the total potassium content with three biological repetitions. The samples were dried using an oven. The finely ground dry samples were digested with sulfuric acid and hydrogen peroxide (5:1, *v*/*v*), and the digestion solution was used to determine the total potassium content using an ICAP6300 ICP spectrometer (Thermo Scientific, Waltham, MA, USA) [58].

### 4.7. Transcriptome Sequencing and Analysis of Differentially Expressed Genes (DEGs)

Similar to the enzyme activity assays, the 14th, 28th, and 42nd days after the first application of potassium-containing fertilizers were selected as time points for transcriptome sequencing with three biological repetitions. The RNA extraction and transcriptome sequencing of the samples were conducted at Beijing Novogene Science and Technology Co., Ltd. (Beijing, China). The RNA of the samples was extracted using RNAprep Pure Plant Kit (Tiangen, Beijing, China) according to the manufacturer’s instructions, and the cDNA libraries were amplified via reverse transcription and PCR. RNA sequencing libraries were then generated using the Hieff NGS Ultima Dual-mode mRNA Library Prep Kit for Illumina (Yeasen Biotechnology Co., Ltd., Shanghai, China) following the manufacturer’s instructions. After sequencing, the raw read data were quality-filtered, and then the sequenced reads were mapped to the reference genome using Hisat2 tools. Subsequently, differential expression analysis between treatments was performed using the DESeq2, and gene ontology (GO) and the Kyoto Encyclopedia of Genes and Genomes (KEGG) enrichment analyses on the resulting DEGs were performed.

### 4.8. Quantitative RT-PCR (qRT-PCR) Analysis of DEGs

The 11 DEGs (*PK*, *PRK*, *SS*, *GolS1*, *PHT14*, *SOR*, *AKT2*, *OCT3*, *CNGC4*, *NFXL1*, and *PT6*) related to sugar and potassium transport that were identified in the transcriptome data were verified by qRT-PCR with three biological repetitions. The primers were designed using Primer v.5.0 (Table 4). qRT-PCR of the DEGs was performed on the CFX96TM Real-Time System platform using the 2× M5 HiPer SYBR Premix EsTaq (with Tli RNaseH) kit from Mei5 Biotechnology Co., Ltd. (Beijing, China), with *ACTIN* as the internal reference gene [63]. The relative DEG expression levels were calculated using the 2^−ΔΔCT^ method.

### 4.9. Statistical Analysis

The data were analyzed in triplicate using Duncan’s Multiple Range Test (*p* < 0.05) of one-way analysis of variance (ANOVA) using SPSS v.20.0.0 (IBM, Inc., Armonk, NY, USA).

## 5. Conclusions

Potassium-containing fertilizers increase the activity of sugar metabolism-related enzymes (SS, SPS, AI, and NI) and organic acid metabolism-related enzymes (PEPC, ME, MDH, CS, ICD, and ACO) at all stages of grape fruit development. During the later stages of grape fruit development, potassium-containing fertilizers increase the total soluble solid content and the sugar content of the different sugar fractions (total soluble sugar, reducing sugar, fructose, sucrose, and glucose) and reduce the titratable acid content and organic acid content of the different organic acid fractions (oxalic acid, tartaric acid, L-malic acid, D-malic acid, citric acid, and succinic acid). Potassium sulfate and potassium dihydrogen phosphate are the most effective at increasing the contents of sugar and total potassium and reducing the organic acid content. Transcriptome analysis shows that the differentially expressed genes (DEGs) in grape berry plants after applying potassium-containing fertilizers are mainly involved in pathways associated with fruit quality, notably carbon metabolism, carbon fixation in photosynthetic organisms, glycolysis and gluconeogenesis, and fructose and mannose metabolism pathways. Four DEGs related to sugar metabolism (*PK*, *PFK*, *GOLS1*, and *SS*) and seven DEGs related to the potassium ion uptake transporter (*SOR*, *CNGC4*, *AKT2*, *NFXL1*, *PT6*, *OCT3*, and *PHT14*) are identified. So, potassium sulfate and potassium dihydrogen phosphate are recommended as suitable potassium fertilizers for grape production.

## Figures and Tables

**Figure 1 ijms-25-02828-f001:**
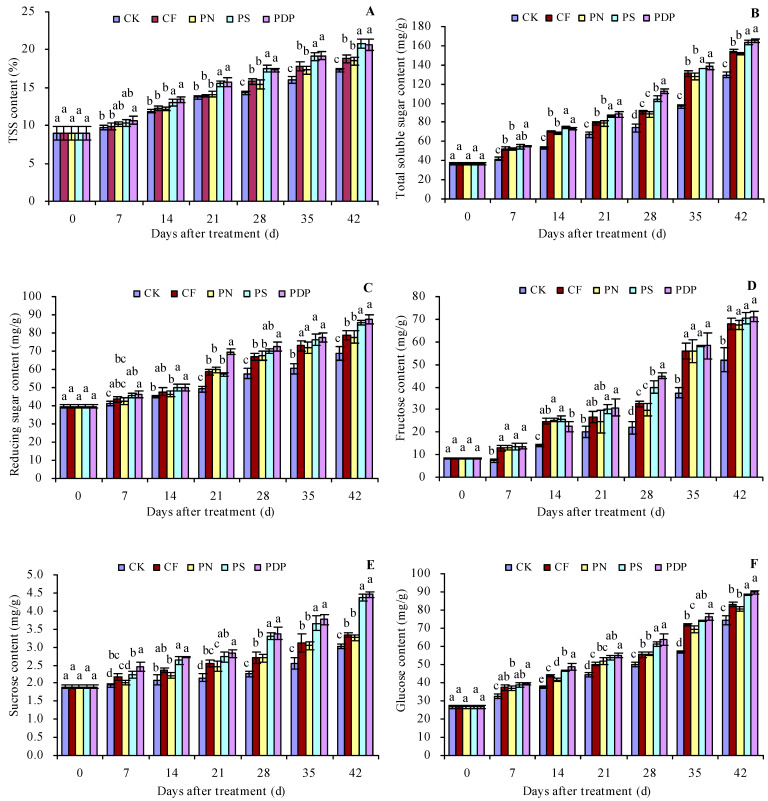
Sugar content in grape fruits. (**A**) Total soluble solid (TSS) content; (**B**) total soluble sugar content; (**C**) reducing sugar content; (**D**) fructose content; (**E**) sucrose content; (**F**) glucose content. Values represent the mean ± SE (n = 3). Different lowercase letters indicate significant differences among the treatments (Duncan’s multiple range test, *p* < 0.05). CK = control; CF = complex fertilizer; PN = potassium nitrate; PS = potassium sulfate; PDP = potassium dihydrogen phosphate.

**Figure 2 ijms-25-02828-f002:**
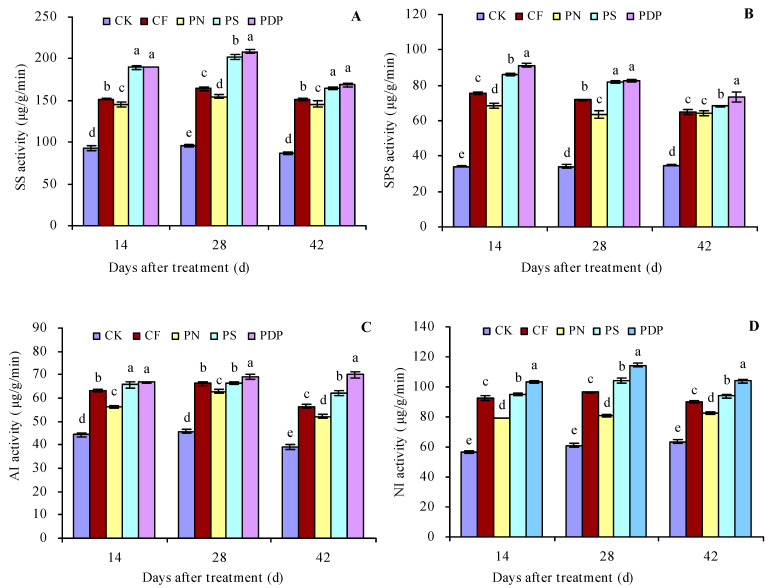
Sugar metabolism-related enzyme activities in grape fruits. (**A**) Sucrose synthase (SS) activity; (**B**) sucrose phosphate synthase (SPS) activity; (**C**) acidic invertase (AI) activity; (**D**) neutral invertase (NI) activity. Values represent the mean ± SE (n = 3). Different lowercase letters indicate significant differences among the treatments (Duncan’s multiple range test, *p* < 0.05). CK = control; CF = complex fertilizer; PN = potassium nitrate; PS = potassium sulfate; PDP = potassium dihydrogen phosphate.

**Figure 3 ijms-25-02828-f003:**
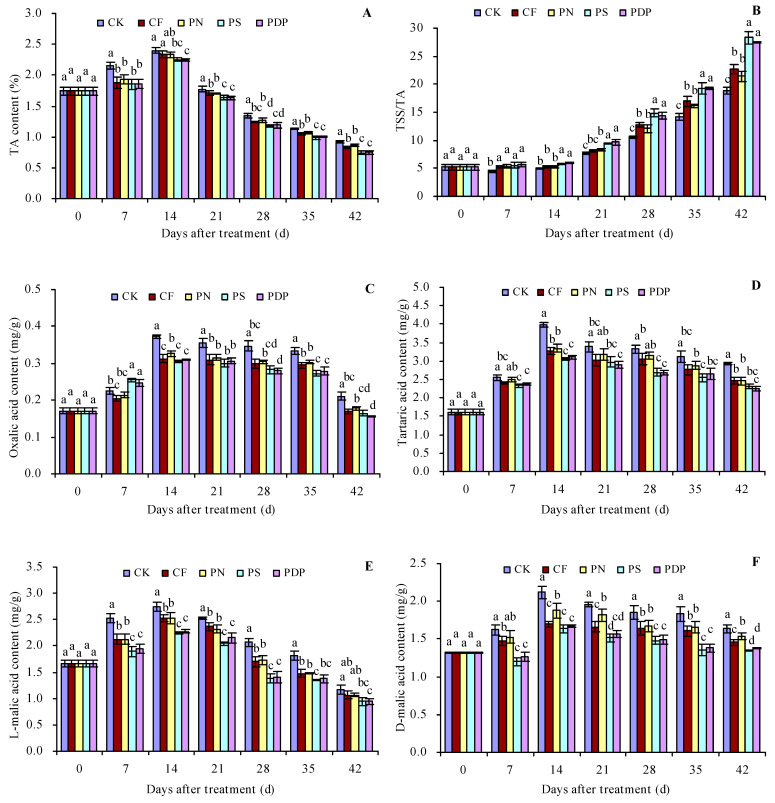
Organic acid content in grape fruits. (**A**) titratable acid (TA) content; (**B**) total soluble solids (TSS) content/titratable acid (TA) content; (**C**) oxalic acid content; (**D**) tartaric acid content; (**E**) L-malic acid content; (**F**) D-malic acid content; (**G**) citric acid content. (**H**) succinic acid content. Values represent the mean ± SE (n = 3). Different lowercase letters indicate significant differences among the treatments (Duncan’s Multiple Range Test, *p* < 0.05). CK = control; CF = complex fertilizer; PN = potassium nitrate; PS = potassium sulfate; PDP = potassium dihydrogen phosphate.

**Figure 4 ijms-25-02828-f004:**
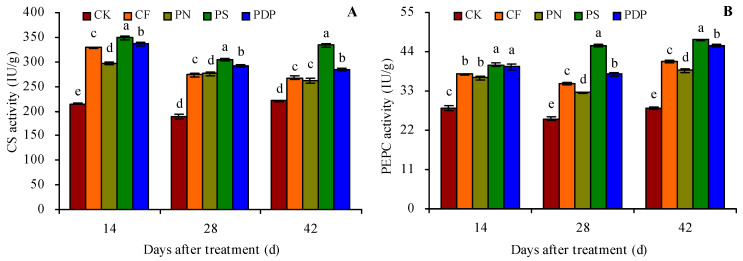
Activities of organic acid metabolism-related enzymes in grape fruits. (**A**) citrate synthetase (CS) activity; (**B**) phosphoenolpyruvate carboxylase (PEPC) activity; (**C**) malic enzyme (ME) activity; (**D**) malate dehydrogenase (MDH) activity; (**E**) aconitase (ACO) activity; (**F**) isocitrate dehydrogenase (ICD) activity. Values represent the mean ± SE (n = 3). Different lowercase letters indicate significant differences among the treatments (Duncan’s Multiple Range Test, *p* < 0.05). CK = control; CF = complex fertilizer; PN = potassium nitrate; PS = potassium sulfate; PDP = potassium dihydrogen phosphate.

**Figure 5 ijms-25-02828-f005:**
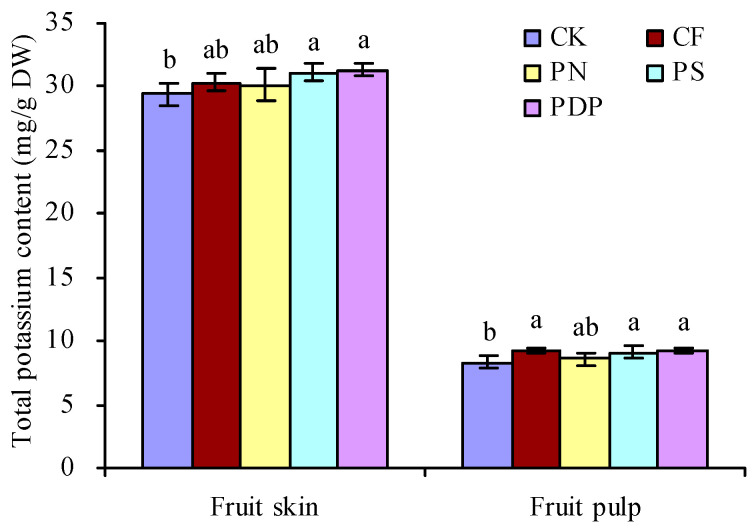
Total potassium content in grape fruits on the 42nd day after treatment. Values represent the mean ± SE (n = 3). Different lowercase letters indicate significant differences among the treatments (Duncan’s multiple range test, *p* < 0.05). DW = dry weight. CK = control; CF = complex fertilizer; PN = potassium nitrate; PS = potassium sulfate; PDP = potassium dihydrogen phosphate.

**Figure 6 ijms-25-02828-f006:**
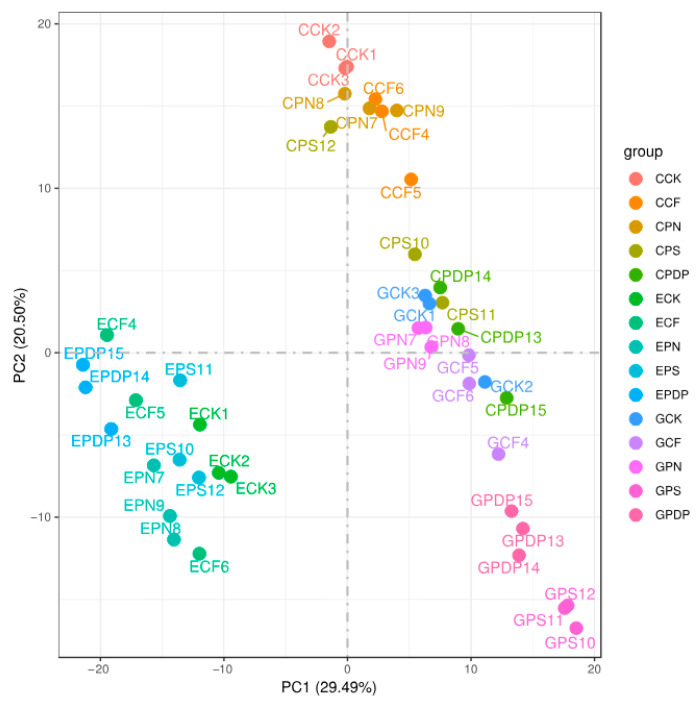
Principal component analysis among treatments. CCK, CCF, CPN, CPS, and CPDP represent the control, complex fertilizer, potassium nitrate, potassium sulfate, and potassium dihydrogen phosphate, respectively, at 14 days after treatment. ECK, ECF, EPN, EPS, and EPDP represent the control, complex fertilizer, potassium nitrate, potassium sulfate, and potassium dihydrogen phosphate, respectively, at 28 days after treatment. GCK, GCF, GPN, GPS, and GPDP represent the control, complex fertilizer, potassium nitrate, potassium sulfate, and potassium dihydrogen phosphate, respectively, at 42 days after treatment.

**Figure 7 ijms-25-02828-f007:**
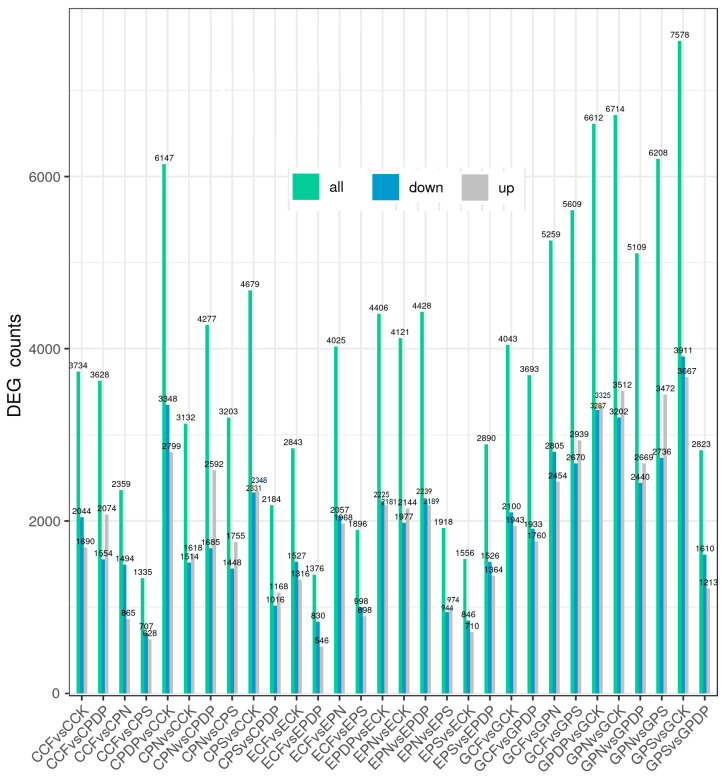
Total numbers of DEGs, upregulated genes, and downregulated genes. CCK, CCF, CPN, CPS, and CPDP represent the control, complex fertilizer, potassium nitrate, potassium sulfate, and potassium dihydrogen phosphate, respectively, at 14 days after treatment. ECK, ECF, EPN, EPS, and EPDP represent the control, complex fertilizer, potassium nitrate, potassium sulfate, and potassium dihydrogen phosphate, respectively, at 28 days after treatment. GCK, GCF, GPN, GPS, and GPDP represent the control, complex fertilizer, potassium nitrate, potassium sulfate, and potassium dihydrogen phosphate, respectively, at 42 days after treatment.

**Figure 8 ijms-25-02828-f008:**
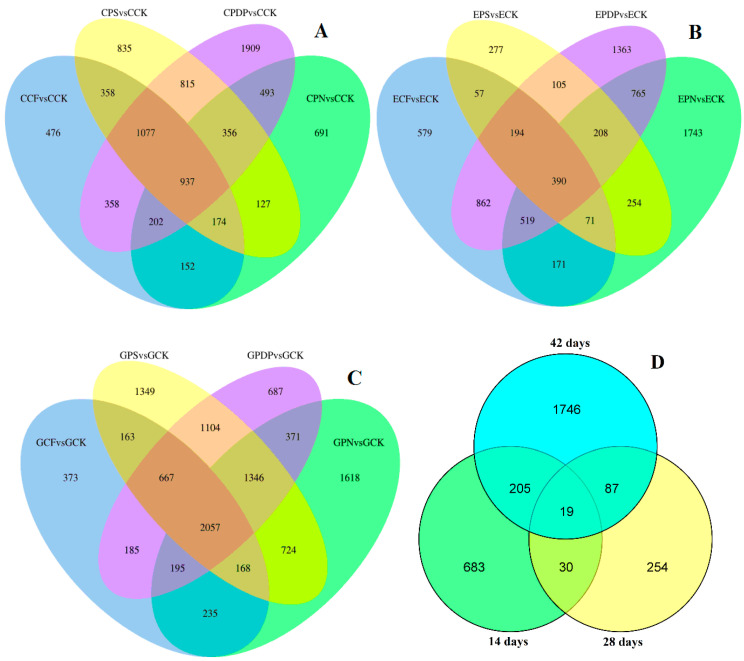
Venn diagram of DEGs. (**A**) DEGs of four treatments at 14 days after treatment; (**B**) DEGs of four treatments at 28 days after treatment; (**C**) DEGs of four treatments at 42 days after treatment. (**D**) DEGs of three periods. CCK, CCF, CPN, CPS, and CPDP represent the control, complex fertilizer, potassium nitrate, potassium sulfate, and potassium dihydrogen phosphate, respectively, at 14 days after treatment. ECK, ECF, EPN, EPS, and EPDP represent the control, complex fertilizer, potassium nitrate, potassium sulfate, and potassium dihydrogen phosphate, respectively, at 28 days after treatment. GCK, GCF, GPN, GPS, and GPDP represent the control, complex fertilizer, potassium nitrate, potassium sulfate, and potassium dihydrogen phosphate, respectively, at 42 days after treatment.

**Figure 9 ijms-25-02828-f009:**
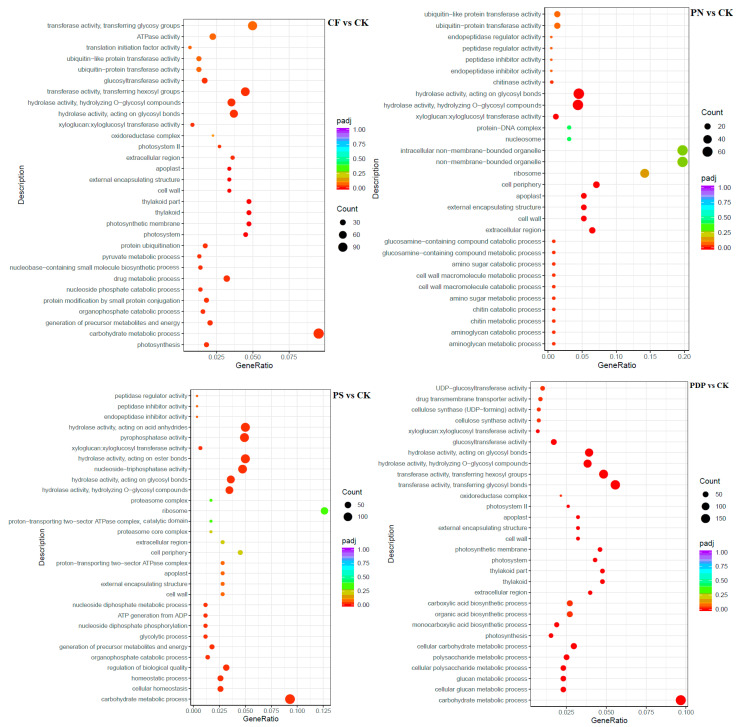
GO enrichment analysis of DEGs at 14 days after treatment. CK = control; CF = complex fertilizer; PN = potassium nitrate; PS = potassium sulfate; PDP = potassium dihydrogen phosphate.

**Figure 10 ijms-25-02828-f010:**
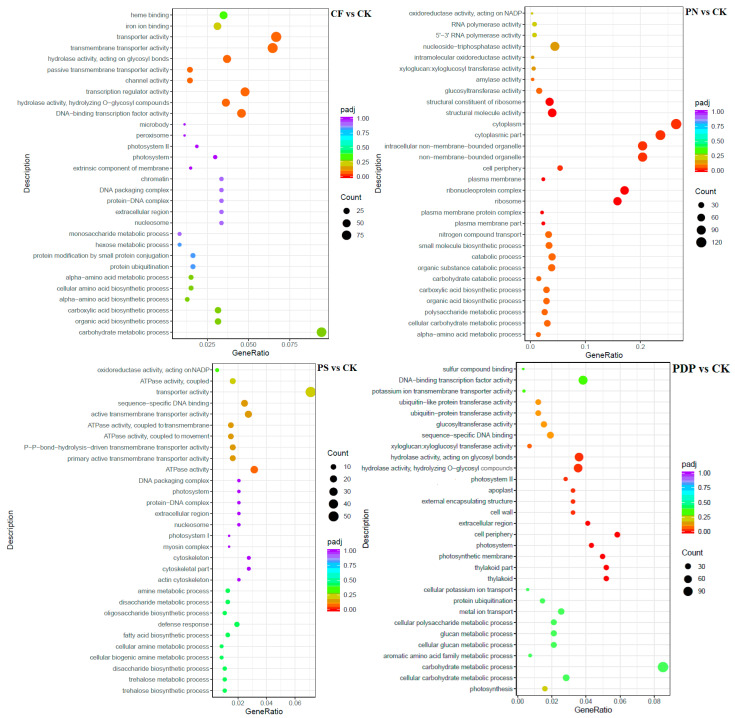
GO enrichment analysis of DEGs at 28 days after treatment. CK = control; CF = complex fertilizer; PN = potassium nitrate; PS = potassium sulfate; PDP = potassium dihydrogen phosphate.

**Figure 11 ijms-25-02828-f011:**
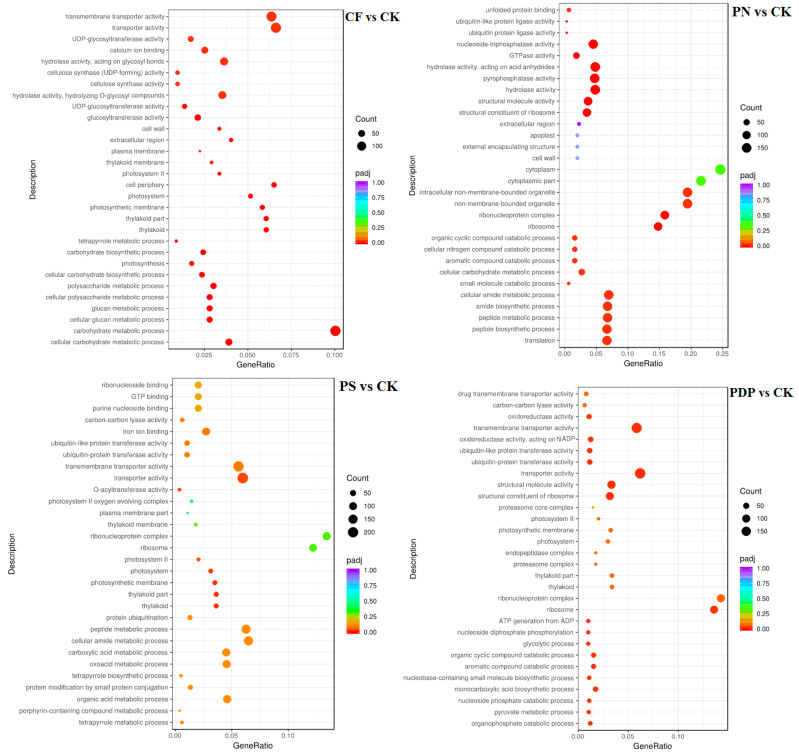
GO enrichment analysis of DEGs at 42 days after treatment. CK = control; CF = complex fertilizer; PN = potassium nitrate; PS = potassium sulfate; PDP = potassium dihydrogen phosphate.

**Figure 12 ijms-25-02828-f012:**
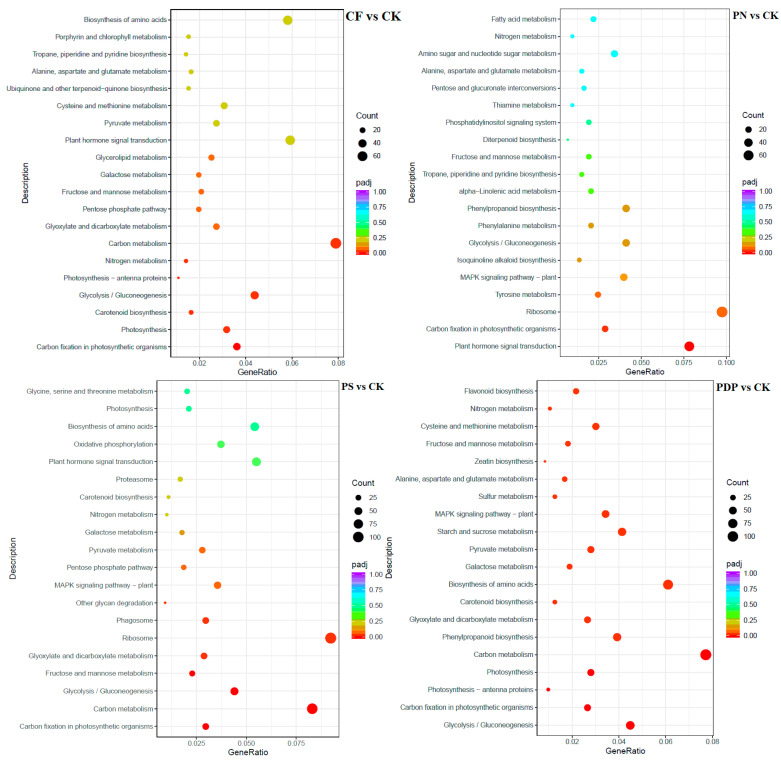
KEGG enrichment analysis of DEGs at 14 days after treatment. CK = control; CF = complex fertilizer; PN = potassium nitrate; PS = potassium sulfate; PDP = potassium dihydrogen phosphate.

**Figure 13 ijms-25-02828-f013:**
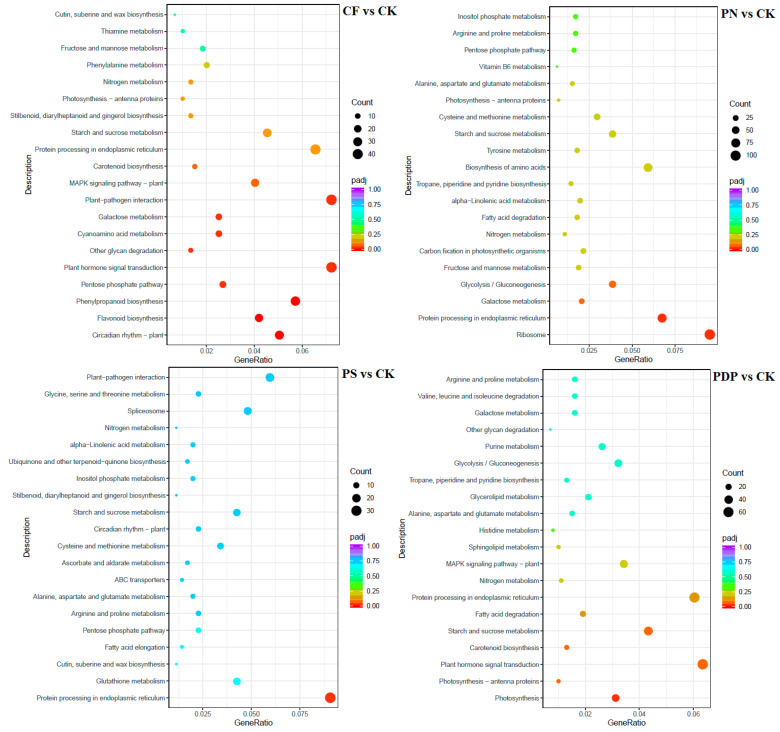
KEGG enrichment analysis of DEGs at 28 days after treatment. CK = control; CF = complex fertilizer; PN = potassium nitrate; PS = potassium sulfate; PDP = potassium dihydrogen phosphate.

**Figure 14 ijms-25-02828-f014:**
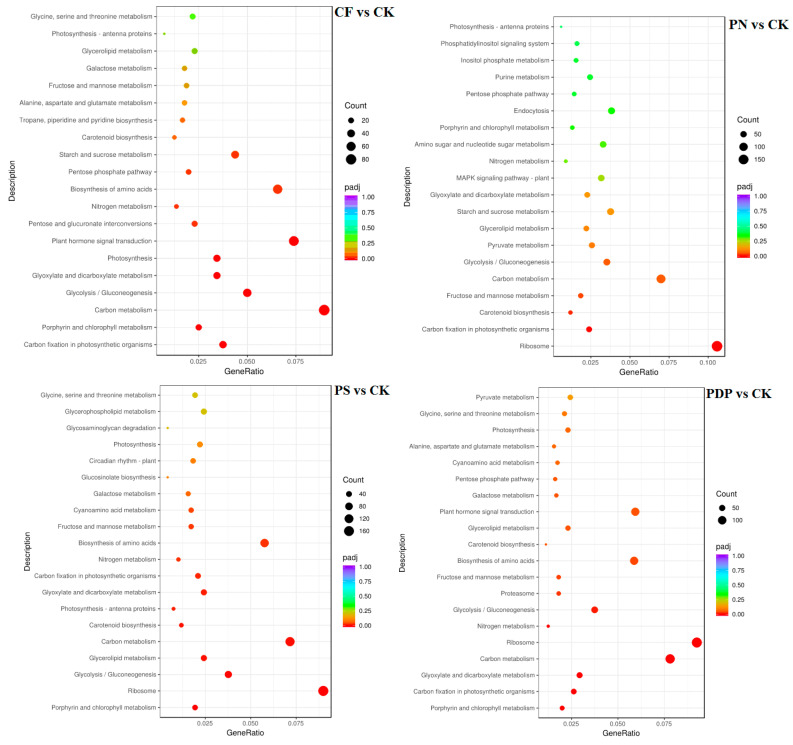
KEGG enrichment analysis of DEGs at 42 days after treatment. CK = control; CF = complex fertilizer; PN = potassium nitrate; PS = potassium sulfate; PDP = potassium dihydrogen phosphate.

**Figure 15 ijms-25-02828-f015:**
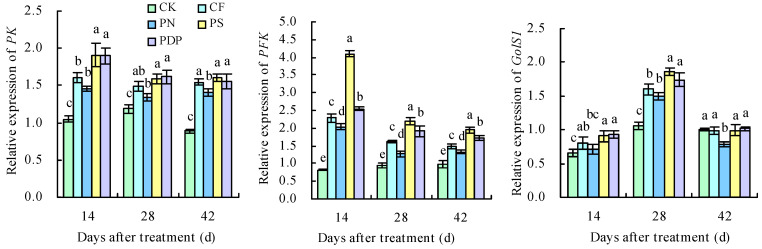
Expression levels of screened DEGs. Different lowercase letters indicate significant differences among the treatments (Duncan’s multiple range test, *p* < 0.05). CK = control; CF = complex fertilizer; PN = potassium nitrate; PS = potassium sulfate; PDP = potassium dihydrogen phosphate.

**Table 1 ijms-25-02828-t001:** Statistical table of transcriptome sequencing data.

Sample	Raw Reads	Clean Reads	Clean Bases	GC Content	Q20 Percentage	Q30 Percentage
CCK	47,431,499	46,848,865	7.03 G	46.59%	97.28%	92.61%
CCF	46,678,397	45,942,010	6.89 G	46.55%	97.44%	93.06%
CPN	47,272,605	46,232,352	6.93 G	46.53%	97.43%	93.02%
CPS	45,467,270	44,788,052	6.72 G	46.59%	97.56%	93.26%
CPDP	50,756,362	50,014,186	7.5 G	46.57%	97.57%	93.34%
ECK	47,031,424	46,243,460	6.94 G	46.42%	97.43%	93.06%
ECF	45,249,647	44,366,584	6.66 G	46.43%	97.71%	93.66%
EPN	48,886,265	48,140,646	7.22 G	46.74%	97.72%	93.62%
EPS	47,141,936	46,461,278	6.97 G	46.52%	97.84%	93.87%
EPDP	48,923,110	48,030,740	7.20 G	46.38%	97.52%	93.13%
GCK	44,575,317	43,594,313	6.54 G	46.43%	96.92%	91.86%
GCF	45,679,846	44,867,686	6.73 G	46.43%	97.38%	92.74%
GPN	47,601,313	46,808,660	7.02 G	46.66%	97.86%	93.90%
GPS	46,993,310	46,268,872	6.94 G	46.45%	97.74%	93.74%
GPDP	45,907,787	44,653,306	6.70 G	47.14%	97.70%	93.73%

CCK, CCF, CPN, CPS, and CPDP represent the control, complex fertilizer, potassium nitrate, potassium sulfate, and potassium dihydrogen phosphate, respectively, at 14 days after treatment. ECK, ECF, EPN, EPS, and EPDP represent the control, complex fertilizer, potassium nitrate, potassium sulfate, and potassium dihydrogen phosphate, respectively, at 28 days after treatment. GCK, GCF, GPN, GPS, and GPDP represent the control, complex fertilizer, potassium nitrate, potassium sulfate, and potassium dihydrogen phosphate, respectively, at 42 days after treatment.

**Table 2 ijms-25-02828-t002:** Overview of KEGG analysis of DEGs.

KEGG ID	Metabolic Pathways	14 Days after Treatment				28 Days after Treatment				42 Days after Treatment				Total Number of DEGs for Pathway Annotations
		CF vs. CK	PN vs. CK	PS vs. CK	PDP vs. CK	CF vs. CK	PN vs. CK	PS vs. CK	PDP vs. CK	CF vs. CK	PN vs. CK	PS vs. CK	PDP vs. CK	
vvi03010	Ribosome	9	71	109	0	0	106	0	0	0	177	167	148	787
vvi01200	Carbon metabolism	72	0	98	110	0	0	0	0	86	0	133	125	624
vvi00860	Porphyrin and chlorophyll metabolism	17	0	0	0	0	0	0	0	24	0	36	32	109
vvi04075	Plant hormone signal transduction	0	57	0	0	0	0	0	0	71	0	0	95	223
vvi00051	Fructose and mannose metabolism	0	0	27	26	0	0	0	0	0	31	32	29	145
vvi00010	Glycolysis/Gluconeogenesis	40	0	52	64	0	0	0	0	48	0	70	80	354
vvi00710	Carbon fixation in photosynthetic organisms	33	21	35	38	0	0	0	0	36	40	39	42	284
vvi01230	Biosynthesis of amino acids	0	0	0	87	0	0	0	0	63	0	107	94	351
vvi00906	Carotenoid biosynthesis	15	0	0	18	0	0	0	0	0	20	22	18	93

**Table 3 ijms-25-02828-t003:** Selected DEGs related to sugar metabolism and potassium transport.

No.	Gene ID in NCBI	Gene Description	*p* Value
1	100255934	*Pyruvate kinase* (*PK*)	0.012
2	100249662	*Phosphofructokinase* (*PFK*)	1.2 × 10^−3^
3	100260266	*Galactinol synthase-1* (*GolS1*)	3.0 × 10^−11^
4	100249279	*Sucrose synthase* (*SS*)	9.3 × 10^−15^
5	100233111	*Shaker-like potassium channel* (*SOR*)	1.0 × 10^−5^
6	100265816	*Cyclic nucleotide-gated ion channel 4* (*CNGC4*)	1.9 × 10^−3^
7	100261806	*Potassium channel AKT2/3* (*AKT2*)	0.013
8	100268010	*NF-X1-type zinc finger protein NFXL1* (*NFXL1*)	6.7 × 10^−6^
9	100251966	*Potassium transporter 6* (*PT6*)	0.013
10	100263786	*Organic cation/carnitine transporter 3* (*OCT3*)	3.2 × 10^−4^
11	100257925	*Low-affinity inorganic phosphate transporter 1* (*PHT14*)	1.4 × 10^−12^

**Table 4 ijms-25-02828-t004:** Primer information of qRT-PCR.

Gene Name	Gene ID in NCBI	Forward Primer Sequence (5′-3′)	Reverse Primer Sequence (5′-3′)
*PK*	100255934	AGGATCAGGTCAACTCCTCT	ATCACAACGATGAGCTTTGC
*PFK*	100249662	TAAGCCGATCCTTTCCTCAC	CTAATGGGTGGATGCTCAGT
*GolS1*	100260266	GCAGATGTGTTTACAGCCG	TTCACATAGTCCCCGTTTCC
*SS*	100249279	TTCAGCCCCGTTGAGAATAA	CAACAAGTCCCGTCAAGTTC
*SOR*	100233111	GCAACCTAACAGCTCCTTTG	CGCAGAAGCCTACACAATTC
*CNGC4*	100265816	AGAGTCAAGAGAAGTGCGAG	TCTTGGCCGGATAAATGACA
*AKT2*	100261806	GAATGGCGCAGATGTTATCC	AATTCGGTGTCCCATCTCTC
*NFXL1*	100268010	TCACGTTTTCTGTCCAATGC	AATAAAACGCTTTGGCTCCC
*PT6*	100251966	CCCGAGAGTCCAAAGATTGA	TTCTTCACCATGCTAGACCC
*OCT3*	100263786	GAGACCGGCTATAAGAGAGC	GAGGAGAGGGGTTGAATCTG
*PHT14*	100257925	GCGTTAATCGACAAGATGGG	AGAGTACATCACCACGAACC
*ACTIN*	100232866	CTTGCATCCCTCAGCACCTT	TCCTGTGGACAATGGATGGA

## Data Availability

The data presented in this study are available upon request from the corresponding author. The data are not publicly available due to privacy reasons.

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
