# Peer review of "Effects of Potassium-Containing Fertilizers on Sugar and Organic Acid Metabolism in Grape Fruits"

_ijms, 2024, doi:10.3390/ijms25052828_

Round 1
Reviewer 1 Report
Comments and Suggestions for Authors
Title
Grape berry fruits - strange wording - it is recommended to use either grape berries or grape fruits, but not both together. It's all over the text.
Keywords. Repeating the same words in the title and keywords is not desirable. This reduces the chance of finding the article in databases and search engines.
Introduction
“Grape berry fruits require enormous amounts of potassium…”- - such a dramatic statement also requires some numerical justification. High content in grape berries, leaves? Big K take out with yield? High doses of K in standard fertilization practices?
“suitable potassium fertilizers for grape production and their mechanism of action have not yet been investigated.” This is not true; it is an exaggerated statement. Potassium fertilizers for grapes have been studied, various scientific articles have been devoted to it. Of course, much remains unexplored.
“The mechanism behind the sugar-increasing and acidity-reducing responses after potassium fertilizer applications in grape berry fruits is also unknown” - Don't make such claims, you yourself have references in this article about studies in this direction. Rather, it should be emphasized that many questions are still unclear and need to be explored.
Materials and Methods
How much K was given to each plant by fertilizer? It would be useful to indicate this so that the reader does not have to do the calculations themselves.
Was the K content in the grapes not determined? This is strange, because knowledge of the K content of berries provided by different fertilizers could help explain the results.
Results
The results section provides a wide range of data and is generally well presented.
Only, each Figure uses a different colour palette for the experimental variants. Is it done purposefully? The reader must relearn them in the next drawing.
Discussion
The discussion is not a literature review. At the moment, it feels like the vast majority of the text is devoted to various other studies, followed by a couple of sentences about your own research. Focus on analysing your own data and how it relates to other authors' data. Otherwise, your research disappears, you have to look for it, when something will be discussed about the obtained interesting results. This is clearly shown in section 4.2. The first 21 lines are devoted to other fruits and vegetables, then only one sentences about this study devoted to grapes. All other sentences end with a reference to the literature, so - research by others.
4.2. “This may be due to the fact that potassium fertilizers promote potassium ion accumulation in the fruits, and the higher potassium ion content reacts with the organic acids to form salts, thereby resulting in a decrease in organic acid content and suggesting that potassium fertilizers can play a role in lowering fruit acidity.” - If there were data on the content of K in berries, it would be possible to conclude convincingly, not just maybe. Perhaps a better understanding of the effects of each fertilizer could also be achieved.
The practical side of the study is lost in the discussion. One of the aims of this study were to screen for suitable potassium fertilizers in grape production. Neither in the discussion nor in the conclusions can I find a clear conclusion or any discussion of this issue. All useful? Which were better? Can the recommended fertilizer be based on the sweetness, acidity of the berries? Which berries are better, for which purposes (juice extraction, winemaking, as dessert berries)? Was there any effect on the vitality of the plants, even if it was visual. It is clear that the study is more devoted to theoretical questions, but still, some answer must be given to the goals set in the study.
In general, the discussion should be more focused on the analysis of the results of your research and only then on their connection with the previous ones. It should also be devoted to all the goals and objectives set in the study.
Conclusions
So what exactly do you recommend as suitable potassium fertilizers for grape production. It can be guessed, but not clearly stated.
I recommend accepting this article in Int. J. Mol. Sci. after minor revision.
Author Response
Comments and Suggestions for Authors
RESPOND: Thank you for your reviewing.
Title
Grape berry fruits - strange wording - it is recommended to use either grape berries or grape fruits, but not both together. It's all over the text.
RESPOND: We have revised as “grape fruits”.
Keywords. Repeating the same words in the title and keywords is not desirable. This reduces the chance of finding the article in databases and search engines.
RESPOND: We have revised as “fruits; potassium; quality; transcriptome analysis”.
Introduction
“Grape berry fruits require enormous amounts of potassium…”- - such a dramatic statement also requires some numerical justification. High content in grape berries, leaves? Big K take out with yield? High doses of K in standard fertilization practices?
RESPOND:
“suitable potassium fertilizers for grape production and their mechanism of action have not yet been investigated.” This is not true; it is an exaggerated statement. Potassium fertilizers for grapes have been studied, various scientific articles have been devoted to it. Of course, much remains unexplored.
RESPOND: We have revised as “the action mechanism of potassium fertilizers on grape fruits still have many questions and need to be explored.”
“The mechanism behind the sugar-increasing and acidity-reducing responses after potassium fertilizer applications in grape berry fruits is also unknown” - Don't make such claims, you yourself have references in this article about studies in this direction. Rather, it should be emphasized that many questions are still unclear and need to be explored.
RESPOND: We have revised as “the action mechanism of potassium fertilizers on grape fruits still have many questions and need to be explored.”
Materials and Methods
How much K was given to each plant by fertilizer? It would be useful to indicate this so that the reader does not have to do the calculations themselves.
RESPOND: It’s 191 g K2O per plant. The corresponding amount of fertilizer that single plant received was: 1272 g of complex fertilizer, 408 g of potassium nitrate, 366 g of potassium sulfate, and 552 g of potassium dihydrogen phosphate. We have added in the text.
Was the K content in the grapes not determined? This is strange, because knowledge of the K content of berries provided by different fertilizers could help explain the results.
RESPOND: We have added the K content in the text.
Results
The results section provides a wide range of data and is generally well presented.
Only, each Figure uses a different colour palette for the experimental variants. Is it done purposefully? The reader must relearn them in the next drawing.
RESPOND: We have revised using the same color for the same treatments in every figure.
Discussion
The discussion is not a literature review. At the moment, it feels like the vast majority of the text is devoted to various other studies, followed by a couple of sentences about your own research. Focus on analysing your own data and how it relates to other authors' data. Otherwise, your research disappears, you have to look for it, when something will be discussed about the obtained interesting results. This is clearly shown in section 4.2. The first 21 lines are devoted to other fruits and vegetables, then only one sentences about this study devoted to grapes. All other sentences end with a reference to the literature, so - research by others.
RESPOND: We have rewritten the literature review.
4.2. “This may be due to the fact that potassium fertilizers promote potassium ion accumulation in the fruits, and the higher potassium ion content reacts with the organic acids to form salts, thereby resulting in a decrease in organic acid content and suggesting that potassium fertilizers can play a role in lowering fruit acidity.” - If there were data on the content of K in berries, it would be possible to conclude convincingly, not just maybe. Perhaps a better understanding of the effects of each fertilizer could also be achieved.
RESPOND: We have added the total K content in grape fruits, and revised the text.
The practical side of the study is lost in the discussion. One of the aims of this study were to screen for suitable potassium fertilizers in grape production. Neither in the discussion nor in the conclusions can I find a clear conclusion or any discussion of this issue. All useful? Which were better? Can the recommended fertilizer be based on the sweetness, acidity of the berries? Which berries are better, for which purposes (juice extraction, winemaking, as dessert berries)? Was there any effect on the vitality of the plants, even if it was visual. It is clear that the study is more devoted to theoretical questions, but still, some answer must be given to the goals set in the study.
In general, the discussion should be more focused on the analysis of the results of your research and only then on their connection with the previous ones. It should also be devoted to all the goals and objectives set in the study.
RESPOND: We have added the conclusions of suitable potassium fertilizers in grape production in Discussion and Conclusion sections.
Conclusions
So what exactly do you recommend as suitable potassium fertilizers for grape production. It can be guessed, but not clearly stated.
RESPOND: We have added.
I recommend accepting this article in Int. J. Mol. Sci. after minor revision.
RESPOND: Thank you again.
Reviewer 2 Report
Comments and Suggestions for Authors
Review on “Effects of Potassium-Containing Fertilizers on Sugar and Organic Acid Metabolism in Grape Berry Fruits”.
The authors examined the effect of four potassium-containing fertilizers on sugar and organic acid metabolism in grape berry fruits. The study had two aims: to screen for suitable potassium fertilizers in grape production and to decipher the mechanism by which potassium fertilizers affect the sugar and organic acid metabolism in grape berry fruits.
The authors found that potassium-containing fertilizers increase the activity of sugar metabolism-related enzymes and organic acid metabolism-related enzymes at all stages of grape berry fruit development. In addition, they stated that the differentially expressed genes in the grape berry plants after applying potassium-containing fertilizers are mainly involved in pathways that relate to fruit quality.
Potassium is one of the most important macro elements in plants’ growth and development as well as in fruit formation and fruit quality. In addition, potassium is involved in many aspects of plant physiology, activates more than 60 enzymes, has a role in photosynthesis and regulation of stomata opening, favors high energy status, etc. It is essential for glucose generation, which is required for root and shoot development, fruit and leaf development, and nutrient absorption. Furthermore, K plays an important function in maintaining cell water content as well as the production and mobilization of carbohydrates in plant tissues. So the function of K in plants’ physiology and fruit quality is well-studied. However, this study may have great importance in grape production because transcriptome analysis is involved in the study.
Generally, I think that the topic of the paper is very interesting and has great importance. Overall the manuscript is clear, organized, and well-structured. The introduction was well-written and gave a good background of the main idea of the study. Results are statistically analyzed. The manuscript contains many valuable results. The research topic fits the aims and topic of the International Journal of Molecular Sciences. However, the manuscript needs improvement before publication. Please check the whole manuscript carefully. Carefully check the manuscript preparation instructions and correct the submitted manuscript (the order of the section should be the following: Abstract, Keywords, Introduction, Results, Discussion, Materials and Methods, and Conclusions).
https://www.mdpi.com/journal/ijms/instructions
Check the superscript in the whole manuscript.
Also carefully check the style of references.
Specific comments:
Introduction:
Please add information related to grape production in China. (growing area, main using forms of grape like juice, fresh consumption, wine, etc., yield t/ha, etc.)
Materials and Methods:
2.5. Determination of the Sugar and Organic Acid Metabolism-Related Enzyme Activity: Please write out the full name of the each enzymes.
Please add the number of repetitions of each measured parameter in the Materials and Methods section.
Results:
Figure 1 and Figure 3 are very crowded, the statistical analysis can’t see easily and clearly.
Please use the same color for the same treatments in every figure.
Author Response
Comments and Suggestions for Authors
Review on “Effects of Potassium-Containing Fertilizers on Sugar and Organic Acid Metabolism in Grape Berry Fruits”.
The authors examined the effect of four potassium-containing fertilizers on sugar and organic acid metabolism in grape berry fruits. The study had two aims: to screen for suitable potassium fertilizers in grape production and to decipher the mechanism by which potassium fertilizers affect the sugar and organic acid metabolism in grape berry fruits.
The authors found that potassium-containing fertilizers increase the activity of sugar metabolism-related enzymes and organic acid metabolism-related enzymes at all stages of grape berry fruit development. In addition, they stated that the differentially expressed genes in the grape berry plants after applying potassium-containing fertilizers are mainly involved in pathways that relate to fruit quality.
Potassium is one of the most important macro elements in plants’ growth and development as well as in fruit formation and fruit quality. In addition, potassium is involved in many aspects of plant physiology, activates more than 60 enzymes, has a role in photosynthesis and regulation of stomata opening, favors high energy status, etc. It is essential for glucose generation, which is required for root and shoot development, fruit and leaf development, and nutrient absorption. Furthermore, K plays an important function in maintaining cell water content as well as the production and mobilization of carbohydrates in plant tissues. So the function of K in plants’ physiology and fruit quality is well-studied. However, this study may have great importance in grape production because transcriptome analysis is involved in the study.
Generally, I think that the topic of the paper is very interesting and has great importance. Overall the manuscript is clear, organized, and well-structured. The introduction was well-written and gave a good background of the main idea of the study. Results are statistically analyzed. The manuscript contains many valuable results. The research topic fits the aims and topic of the International Journal of Molecular Sciences. However, the manuscript needs improvement before publication. Please check the whole manuscript carefully. Carefully check the manuscript preparation instructions and correct the submitted manuscript (the order of the section should be the following: Abstract, Keywords, Introduction, Results, Discussion, Materials and Methods, and Conclusions).
https://www.mdpi.com/journal/ijms/instructions
Check the superscript in the whole manuscript.
Also carefully check the style of references.
RESPOND: Thank you for your reviewing. We have checked and revised the structure and format of manuscript.
Specific comments:
Introduction:
Please add information related to grape production in China. (growing area, main using forms of grape like juice, fresh consumption, wine, etc., yield t/ha, etc.)
RESPOND: We have added as “In 2021, the cultivation area of grape in China is 783,000 ha, ranking third in the world, and the grape yield is 11.2 million tons, with the yield of table grapes accounting for 90% of the world's production.”
Materials and Methods:
2.5. Determination of the Sugar and Organic Acid Metabolism-Related Enzyme Activity: Please write out the full name of the each enzymes.
RESPOND: We have added.
Please add the number of repetitions of each measured parameter in the Materials and Methods section.
RESPOND: We have added.
Results:
Figure 1 and Figure 3 are very crowded, the statistical analysis can’t see easily and clearly.
RESPOND: We have revised.
Please use the same color for the same treatments in every figure.
RESPOND: We have revised.